# Design, Implementation, and Evaluation of Healthcare Visual Map Tool for Health Workers to Improve Quality of Life of Home Care Patients: Study Protocol

**DOI:** 10.3390/healthcare13060626

**Published:** 2025-03-13

**Authors:** Núria Hernández-Vidal, Marta Pujol-Vidal, Yolanda Mengíbar-García, David Ayala-Villuendas, Joan-Carlos Contel-Segura, Sara Martínez-Torres, Anna Bordas, Eulalia Oriol-Colominas, Nuria Martín-Vergara, Francisco Martín-Lujan, Maria-Pilar Astier-Peña, Montserrat Gens-Barberà

**Affiliations:** 1Quality and Patient Safety Central Functional Unit, Gerència d’Atenció Primària Camp de Tarragona, Institut Català de la Salut, 43005 Tarragona, Spain; nhernandez.tgn.ics@gencat.cat (N.H.-V.); mpujolv.tgn.ics@gencat.cat (M.P.-V.); ymengibar.tgn.ics@gencat.cat (Y.M.-G.); eoriol.tgn.ics@gencat.cat (E.O.-C.); 2QiSP-Tar Research Group, Fundació Institut Universitari per a la Recerca a l’Atenció Primària de Salut—IDIAPJGol, 08007 Barcelona, Spain; dayala.tgn.ics@gencat.cat (D.A.-V.); smartinez@idiapjgol.org (S.M.-T.); abordas.tgn.ics@gencat.cat (A.B.); fmartin.tgn.ics@gencat.cat (F.M.-L.); mpastier@gmail.com (M.-P.A.-P.); 3ISAC Research Group (Intervencions Sanitàries i Activitats Comunitàries, 2021 SGR 00884), Fundació Institut Universitari per a la Recerca a l’Atenció Primària de Salut—IDIAPJGol, 08007 Barcelona, Spain; 4Technology and Informatics Department Camp de Tarragona, Institut Català de la Salut, 43005 Tarragona, Spain; 5Chronic and Integrated Care Unit, Department of Health, 08028 Barcelona, Spain; jccontel@gencat.cat; 6Research Support Unit Tarragona-Reus, Direcció d’Atenció Primària Camp de Tarragona, Institut Català de la Salut, Fundació Institut Universitari per a la Recerca a l’Atenció Primària de Salut—IDIAPJGol, 43202 Reus, Spain; nmartin.tgn.ics@gencat.cat; 7Department of Medicine and Surgery, School of Medicine and Health Sciences, Universitat Rovira i Virgili, 43201 Reus, Spain; 8Universitas Health Center, Health Service of Aragon, 50017 Zaragoza, Spain

**Keywords:** patient safety, quality of healthcare, information technology, primary healthcare, home care service

## Abstract

**Background:** Home care programs in primary healthcare aim to address the health needs of elderly and frail populations, but ensuring safety in home environments can be challenging due to factors like limited supervision, environmental hazards, and the complexity of managing multiple health conditions. Fragmented information further complicates healthcare professionals’ ability to identify critical health risks and manage them effectively. **Objective:** This study aims to design and evaluate the effectiveness of a visual map integrated into the electronic medical record of home care patients. The goal will be to implement this tool and enhance healthcare professionals’ interventions, improving patient safety and quality of life. **Methods:** The study will be conducted in three phases: (1) the design of a visual map for the home care program by identifying health areas and contributing factors; (2) a cluster-randomized clinical trial across primary care centers to assess whether the visual map reduces hospital admissions, falls, and injuries while improving quality of life; and (3) a qualitative study to evaluate the impact of the visual map on the home care program. **Results:** It is anticipated that the visual map will reduce hospital admissions, falls, and injuries among home care patients while improving quality of life by facilitating targeted and effective professional interventions. **Discussion:** This visual map aims to enhance the safety of vulnerable home care patients by integrating critical health information into a practical, easy-to-implement tool. Its feasibility and potential for integration into routine primary care within the Catalan Institute of Health make it a valuable innovation with broader applicability. Trial registration: NCT04399616. Registered on 8 May 2024.

## 1. Introduction

In recent decades, rising life expectancy has contributed has contributed to a growing aging population in developed countries. In 2022, life expectancy at birth in Europe was 80.6 years, whereas in Spain, it reached 83.2 years [1]. Currently, nearly 20% of the Spanish population is aged 65 or older, a percentage that is projected to rise to 26% by 2037 [2]. Among this population, approximately 70% have chronic conditions, 26.3% have a disability, and 14% are dependent. Additionally, by the age of 75, more than half of the population (57.8%) experience disabilities and nearly 40% (38.5%) require assistance for daily activities, making their care a pressing challenge for healthcare systems [3].

Frailty, as defined in the “Conceptual bases and care model for frail people with complex chronicity or advanced chronicity” guides, is a multidimensional state of “vulnerability” to stressful factors. This occurs due to the limitation of compensatory mechanisms, putting the individual at a high risk of adverse health outcomes, such as disability, institutionalization, hospitalization, prolonged hospital stays, readmissions, falls, adverse reactions resulting from some specific interventions (for example, chemotherapy or surgeries), and, especially, increased mortality [4].

Managing the health and social needs of frail individuals is complex, requiring coordinated, patient-centered approaches. Consequently, home care programs have emerged as a solution, providing healthcare services in the patient’s home to enhance autonomy and improve quality of life [5].

Home care services, primarily delivered by home care nurses (HCNs), offer high-quality, patient-centered, and cost-effective care. A systematic review, which included a total of nine studies and involved a total of 46,154 individuals aged 65 and older, showed that home care programs reduce healthcare utilization and costs and improve both patient and caregiver satisfaction. Moreover, the review underscored the importance of interdisciplinary teams in coordinating complex patient care as a key driver of improved health outcomes [6]. However, the home environment presents unique challenges, necessitating active collaboration between patients, caregivers, and healthcare professionals [7]. Reliable and accurate information is essential for effective care planning, particularly in areas such as infection risk management [8].

Home care patients are more vulnerable to adverse events (AEs) due to conditions such as mental and physical deterioration, sensory deficits, polymedication, and a lack of professional or caregiver supervision. Despite this vulnerability, home remains the preferred setting for care, emphasizing the need for protocols to ensure patient safety [9].

A study conducted in Sweden found an AE incidence of 37.7%, with 76% considered preventable [10]. A review including studies comprising a pool of 168 articles from several countries reported AE rates ranging from 3.5% to 15.1% [11], similarly to a Canadian cohort of 1200 charts of clients from publicly funded home care programs in the 2009–2010 registry, which indicated an incidence of 4.2%, with 56% being preventable [12]. In all these studies, common AEs in home healthcare include falls, infections, behavioral issues, and medication-related complications [10,11,12]. According to AEs, a recent study concluded that home care patients have an increased likelihood of reporting potentially preventable AEs, particularly drug interactions [13].

Various interventions, such as exercise programs and home safety improvements, have been suggested to prevent AEs, particularly to reduce falls, and significantly improve mobility and activities of daily living in frail older adults [14,15]. Moreover, a Canadian study highlighted communication and team coordination as key factors that could ameliorate the contributing factors to AEs [16].

International initiatives, such as the “European Scaling-up Strategy in Active and Healthy Aging” initiative created by the European Commission, have aimed at the prevention and early diagnosis of frailty and functional decline [17]. For example, a community nursing program in Barcelona reduced healthcare utilization and mortality among older adults by promoting physical activity, proper nutrition, and medication management [18].

The population that is elderly and frail, or at risk of becoming so, need advice focused on their circumstances from HCNs with the aim of preserving their autonomy, improving their quality of life, avoiding AEs, and reducing mental and physical deterioration. By identifying certain factors, proactive actions could be taken to prevent or reduce frailty and improve health and quality of life [9,19]. In Germany, an intervention based on nurse and physician training and the use of a software tool aimed to reduce adverse drug events in home care patients [20]. In Canada, integrative software was used to support person-centered care, clinical and management processes, and proactive care in the home care environment [21].

Despite the availability of extensive patient information in medical records, covering risk areas such as falls, infections, medication safety, caregiver support, autonomy, symptom control, and social determinants, this information remains fragmented across different records. This fragmentation hinders HCNs from quickly identifying critical health concerns and prioritizing patient safety. However, there is a notable gap in the literature regarding the use of information and communication technologies to streamline patient risk identification in home care settings.

To address this issue, the present study leverages information and communication technologies to develop a visual alert map that consolidates key patient data. By translating medical records into an accessible format, this tool aims to enhance HCN practices, facilitate prioritization, and reduce AEs in home care.

## 2. Materials and Methods

### 2.1. Aim of Study

This project’s overall aim is to design, implement, and evaluate the effectiveness of a visual map available in the EMRs of home care patients with the goal of enhancing HCN interventions to improve their safety and quality of life.

The specific objectives of the study are as follows:Develop a visual map to be integrated into EMRs, ensuring that it effectively organizes and prioritizes key patient health information.Assess the effectiveness of the visual map in improving patient safety and care coordination through a cluster-randomized clinical trial conducted in primary care centers in the countryside of Tarragona, Spain.Explore healthcare professionals’ perceptions of the visual map’s impact on home care practices through a qualitative study, identifying strengths, challenges, and areas for improvement.

### 2.2. Setting of the Study

Patients at primary care centers in Tarragona (Spain) that belong to the Catalan Health Institute will be included in the home care program.

This area is represented by 20 primary care team, distributed in 20 primary care centers and more than 70 local clinics, serving a home care population of 3244 users.

### 2.3. Study Development

The general framework of this study is shown in Figure 1. In general, the study includes three phases, each corresponding to a specific objective:Phase (1): the design of the visual map included in the home care program.Phase (2): a cluster-randomized clinical trial across different primary care centers in the countryside of Tarragona, Spain.Phase (3): a qualitative study to evaluate the impact of the visual map’s implementation in the home care program by healthcare professionals.

#### 2.3.1. Phase 1: Design of Home Care Visual Map

The first phase consists of designing a tool based on a visual map available in the EMRs of home care patients.

In this way, the professionals who attend to home care patients will have immediate access to the visual map including the main health areas related to each patient. This phase includes two sub-phases: the identification of health areas and their contributing factors.

Identification of health areas and their contributing factors

First, a group of experts specializing in home care programs and chronic health conditions will be formed. This group will search and discuss the variables available in EMRs that can act as risk factors and classify them into different health areas.

The selection of the variables will be based on data analysis, discussion, and consensus among the experts. The agreed health areas will constitute the visual map. An example of a visual map with health areas and their contributing factors is shown in Figure 2.

2.Risk weighting with a group of experts

Once the health areas and their contributing factors are identified, the risk or problem per area will be weighted as follows: no risk (in green), potential risk (in orange), and current problem (in red). This weighting will be take into account the magnitude or severity of the measured problem. An example of the visual map with risk weighting is shown in Figure 3.

The research team will form a group of experts for each area and the cut-off points will be decided to classify the risk.

Finally, all this information will be registered and integrated into EMRs.

#### 2.3.2. Phase 2: Cluster-Randomized Clinical Trial Across Different Primary Care Centers in Countryside of Tarragona, Spain

A cluster-randomized clinical trial will be carried out for 12 months. The study will be conducted in 20 primary care centers in the Tarragona area. These centers will be randomly allocated into two groups at a ratio of 1:1. The intervention group will have the home care visual map available. No blinding will be implemented in this study. This open-label design is chosen due to the nature of the interventions, which will make blinding unfeasible

This phase of this study will follow the SPIRIT Statement guidelines for reporting clinical trial protocols.

##### Sample Size

The expected effect size is a 10% reduction in hospital admissions, based on the baseline hospital admission rate of 65.89% in 2024 for the home care program at primary care centers in Tarragona, which belong to the Catalan Health Institute. We assumed a 95% confidence level (alpha error of 5%) and a statistical power of 80%. Additionally, a 15% loss to follow-up was factored into the calculation. With these assumptions, a total of 272 patients per group (control and intervention) will be required. The ARCSINUS approximation (GRANMO Sample Size Calculator v.8.0, available at www.datarus.eu/aplicaciones/granmo/) was used.

##### Recruitment

All patients enrolled in the home care program will be eligible for inclusion in this study. Each patient will receive a brief explanation of the study from the responsible investigator at each center, who will be a healthcare professional. The patients will then be invited to sign an informed consent form. In cases where the patient has significant cognitive impairments, their legal guardian or tutor will be asked to provide consent on their behalf. Patients who do not provide consent will be excluded from this study. This process will ensure clear communication, ethical recruitment, and adherence to consent protocols.

There will be no additional exclusion criteria, as this will be a pragmatic study that will aim to represent the reality of the home care population.

Additionally, we will ensure that the recruitment process reflects the representation of the home care population and that key variables—such as sex, degree of dependency, and socio-economic level—are distributed in a way that is representative of the broader patient population.

##### Variables

The main outcome variables of this study are as follows:The proportion of patients with hospital admissions and avoidable hospital emergency visits (main variable) during the follow-up period.The percentage of patients with ulcers and falls.The total number of avoidable admissions and hospital emergency visits during the follow-up period.The total number of deaths and definitive institutionalizations.Quality of life, measured using the EuroQol-5D index, a record currently available in EMRs [22].

The following are considered as covariates:Age, sex, and socio-economic level [23] and the home care model.Main comorbidities, following the Charlson comorbidity index [24].

##### Data Collection and Sources of Information

Most of the data will be collected from EMRs, such as socio-demographic variables and scores on the different scales. Other sources of information will be direct observation, interviews with patients and caregivers, and exchange of information with other HCNs involved, such as HCNs from social services or other healthcare facilities.

To promote participant retention and complete follow-up, monthly sessions will be held to assess the progress of this study. During these sessions, the study’s advancement will be discussed, and any concerns or deviations from the intervention protocol will be addressed.

Study data will be collected and managed using REDCap electronic data-capture tools hosted at Camp de Tarragona of the Catalan Institute of Health [25,26]. REDCap (Research Electronic Data Capture) is a secure, web-based software platform designed to support data capture for research studies, providing (1) an intuitive interface for validated data capture; (2) audit trails for tracking data manipulation and export procedures; (3) automated export procedures for seamless data downloads to common statistical packages; and (4) procedures for data integration and interoperability with external sources.

##### Data Analysis

Continuum variables will be described using means and standard deviations if they follow a normal distribution or using means and interquartile ranges otherwise. The normality of the variables will be described using the Kolmogorov test. Categorical variables will be described with percentages and 95% confidence intervals.

Outcomes will be compared between groups using the Chi-square test. A multivariate analysis (Cox regression) will be performed, introducing adjustment variables into the model: socio-demographic variables, risk areas, and covariates, as mentioned before. Covariate selection will be based on clinical relevance and statistical significance (*p* < 0.05 in univariate analysis or established predictors from previous research). The time until the event of the main outcome variables will be considered in the model.

For the “quality of life” outcome, the test scores (averages and percentiles) will be compared between the control and intervention groups at the beginning and at the end of the intervention using the *t* test.

Missing data will be handled under the principle of the worst-case scenario. However, different imputation strategies may be considered depending on the context.

The statistical package R (R foundation for statistical computing, Vienna, Austria; version 4.1.2 or later) will be used for all analyses. Statistical significance shall be established at *p* < 0.05.

#### 2.3.3. Phase 3: Qualitative Study

##### Design

A qualitative study through semi-structured interviews and focus groups will be performed. The objective of this phase will be to assess the impact of the use of the home care visual map on the healthcare professionals who attend home care patients. This study will assess the usability and effectiveness perceived by the healthcare professionals (whether it facilitates the identification of patient problems, prioritization, and decision-making in healthcare practice and if it is valued as a beneficial tool).

##### Participants

The selection process of the participating healthcare professionals will be carried out based on the specific parameters of this study and aim at finding the maximum variability, heterogeneity, and significance with similar representation by the collective group (medicine, nursing, social work), place of work (rural, urban), and home care model (home care team or basic units). A flexible sampling design will be conceived that also accommodates a circular view of this research and where reformulations can take place based on the results that are obtained throughout the fieldwork.

##### Data Collection and Resources

Data collection will be performed through conversational or narrative techniques, although observation and documentary analysis will also be used.

For the conversational or narrative techniques, the research team will realize semi-structured interviews and focus groups (both with a thematic script) that will allow the collection of information and contextualization, as well as the triangulation of information.

To assess the impact of the visual map implementation in improving the healthcare practice of healthcare professionals, the professional risk category (nurse, social health worker, and physician) and years of experience in primary care will be taken into account.

##### Data Analysis

Data from the interviews and focus groups will be comprehensively systematized with the ATLAS-TI program. The data will be analyzed using thematic contest analysis for its flexibility and suitability in identifying patterns across qualitative data, aligning with our study’s objective of exploring participants’ experiences [27]. Moreover, thematic analysis will allow for a rich and detailed yet complex account of data [28]. At least two researchers will conduct independent analysis and coding. Once the recordings of the interviews and focus groups are transcribed, the researchers will become familiar with the data. The next step will be (1) the identification of relevant topics in the text, (2) the division of the text into units of meaning, (3) coding the data, (4) the generation of categories by grouping codes, (5) the analysis of each category, and (6) the elaboration of new texts.

The research team will discuss the results until a consensus is reached (data triangulation).

##### Data Storage and Dissemination

All data from this study will be stored in a secure database, stripped of identifiers. Methods for data management and coding will be accessible by contacting the corresponding author. The results of this study will be disseminated through publications, reports, and conference presentations.

## 3. Expected Results

This study is expected to generate valuable insights across its three phases, contributing to the improvement of home care for patients through the implementation of a visual map within EMRs.

Phase 1: Development of Home Care Visual Map

By the end of this phase, we expect to have a fully developed and integrated visual map within EMRs. This tool will be rigorously validated during the subsequent clinical trial in phase 2, ensuring its effectiveness and practical applicability in real-world settings. 

Phase 2: A clinical trial to assess the effectiveness of the Visual Map

We anticipate that the trial will confirm several benefits of integrating the visual map into clinical practice. Specifically, we expect to observe a reduction in hospital admissions and emergency visits, as well as a decrease in falls and pressure ulcers among home care patients. Additionally, we hope to see an improvement in overall patient well-being, as measured by the EuroQol-5D index. The findings will also provide valuable data on mortality and institutionalization rates, offering further insight into the long-term impact of the intervention.

The results will be systematically reported, including a breakdown of study participants, their distribution between the intervention and control groups, and a detailed analysis of changes in health outcomes over 12 months. A CONSORT diagram will illustrate the patient flow throughout this study, ensuring transparency in the reporting.

Phase 3: Understanding the experience of healthcare professionals

Healthcare professionals will provide insights into how the visual map integrates into their workflow, whether it facilitates the identification of patient problems, and how it supports collaboration between different disciplines. We expect to gain insights into how the tool fits within existing clinical routines, improves patient problem identification, and enhances communication among teams. Additionally, we aim to identify barriers and facilitators to its implementation. By the end of this study, we anticipate demonstrating the potential of the visual map to enhance home care services, improve patient outcomes, and support healthcare professionals in delivering more efficient and targeted care. 

## 4. Discussion

The adoption of home care programs has significantly increased in response to the growing prevalence of aging and frailty [29]. However, the current system is incapable of identifying the vulnerable areas that are specific to each individual under care. This project aims to introduce a proactive care tool to identify critical health areas for each home care patient.

The primary outcome of this study will be a visual mapping tool incorporating data from the EMRs of patients in home care programs. Integrating this tool into the practice of healthcare professionals can enhance proactive healthcare, enabling the early identification of potential health risks and facilitating early intervention and the application of preventive measures. In addition, it will facilitate the workflow of healthcare professionals, as it will provide real-time and accurate information about a person’s condition, helping in decision-making.

Evaluating the effectiveness of digital technologies on home care patients requires assessing their impact on health and daily functioning [30]. Our study will evaluate the effectiveness of the visual map in reducing hospital admissions and avoidable emergency visits, aligning with the growing body of research that highlights the positive influence of digital tools in improving patient safety and clinical outcomes. Previous studies have demonstrated the positive impact of clinical decision-support systems in clinical practice in the past decade [31]. The decrease in consultation time and misdiagnoses in primary care settings supports the cost-effectiveness of implementing these systems [32]. However, these systems are disease-specific and do not address all the clinical needs of home care patients. Moreover, healthcare professionals have reported experiencing “alert fatigue” and emphasized the need to optimize these systems [33]. Our study goes a step further by focusing specifically on the home care setting, where challenges such as fragmented patient data and a lack of immediate access to clinical information often complicate decision-making.

The visual map will enable healthcare professionals to determine appropriate actions based on the color-coded health areas, whether by implementing health promotion activities, undertaking preventive measures, or addressing existing issues requiring intensive intervention. Moreover, having a standardized visual map for all healthcare professionals will ensure continuity in care and enhance the monitoring of home care patient performance [34].

Despite its advantages, the visual map will have some limitations, including potential technological complexities in implementing EMRs. These challenges may involve integration with existing systems, data accuracy, and the adaptation of the tool to the specific needs of various healthcare settings. Moreover, organizational barriers, such as workflow disruptions and resistance from healthcare professionals, may also affect the tool’s adoption. However, the research team has extensive experience in innovation and the implementation of various technological tools in the field of quality and patient safety [35,36]. In fact, some of these innovations have been successfully scaled up and integrated across the entire autonomous community. To further support the implementation of the visual map, a training session will be conducted at the start of this study, and monthly meetings will be held to monitor usage, address concerns, and provide ongoing support to professionals. This experience and structured support plan will help mitigate the challenges and facilitate the smooth adoption of the tool.

In phase 2 of this study, different socio-demographic and socio-economic characteristics of the patients enrolled could influence the results. To mitigate these differences, similar centers will be selected and a comparative descriptive baseline analysis between the control and intervention groups will be conducted. This approach will help reduce potential bias and improve the robustness of the findings.

Incorporating phase 3 in this study will be crucial for gathering qualitative user feedback on the visual map to ensure that it fits well within the work environment and is intuitive to use. This adaptation aims to increase the use of the visual map in daily routines, ultimately leading to time savings for HCNs [37].

In conclusion, the visual map will enhance the safety of vulnerable patients in a less controllable environment, promoting patient safety, and aiding in resource management and distribution decision-making.

Finally, the analysis of the effectiveness of the visual map will provide initial evidence regarding its potential integration into routine primary care practice within the Catalan Institute of Health in Catalonia, Spain. However, further studies will be necessary to confirm its long-term effectiveness and feasibility in broader clinical settings.

## 5. Conclusions

This study aims to develop and evaluate the implementation of a visual map within EMRs to enhance the management of home care patients. By providing healthcare professionals with a structured and accessible overview of key health areas, this tool will have the potential to improve decision-making, optimize resource allocation, and ultimately enhance patient outcomes.

Through a phased approach, the study will first establish a consensus on relevant health indicators, integrate the visual map into clinical practice, and assess its effectiveness in reducing hospital admissions, falls, and other adverse events. Additionally, qualitative insights from healthcare professionals will provide a deeper understanding of its usability and impact on workflow.

While the findings of this study will offer initial evidence on the benefits of the visual map, further research will be necessary to validate its long-term effectiveness and scalability in broader healthcare settings. The results will contribute to the ongoing improvement of home care services, with the potential for integration into routine practice across the Catalan Institute of Health and also in healthcare systems globally, enhancing the quality of home care services and patient outcomes on a larger scale.

## Figures and Tables

**Figure 1 healthcare-13-00626-f001:**
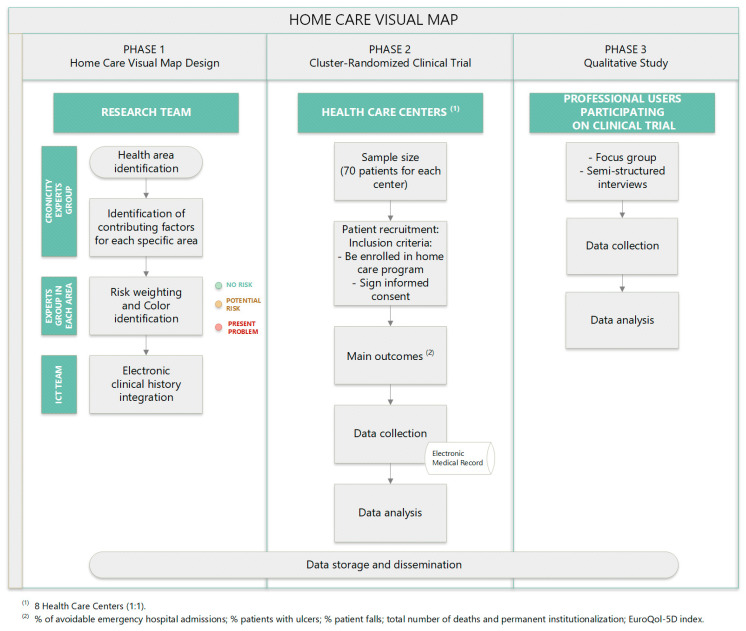
Flowchart with different phases and steps of methodology.

**Figure 2 healthcare-13-00626-f002:**
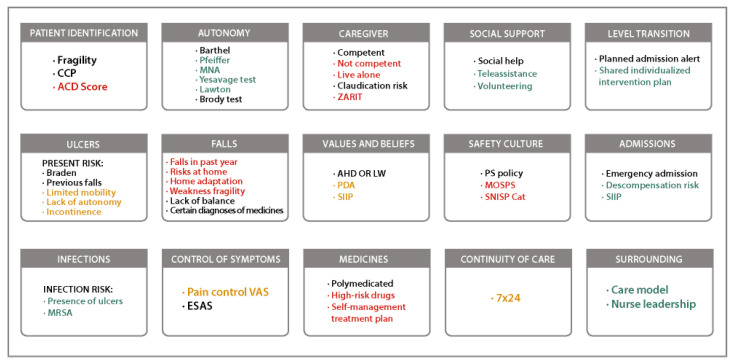
An example of the risk level and weighting map. Contributing factors from the different health areas are colored depending on the risk level: no risk (in green), potential risk (in orange), and current problem (in red). ACD, Advanced Chronic Disease; AHD, Advance Healthcare Directive; CCP, Complex Chronic Patient; ESAS, Edmonton Symptom Assessment System; LW, Living Will; MNA, Mini Nutritional Assessment; MOSPS, Medical Office Survey on Patient Safety; MRSA, Methicillin-resistant Staphylococcus aureus; PDA, Planning Decisions in Advance; PS, patient safety; SIIP, Shared Individualized Intervention Plan; SNISP Cat, Sistema de Notificació d’Incidents de Seguretat dels Pacients de Catalunya; VAS, Visual Analog Scale.

**Figure 3 healthcare-13-00626-f003:**
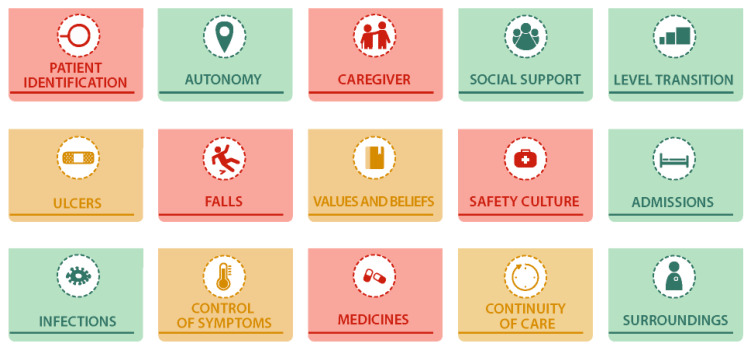
An example of the healthcare visual map. Each health area is colored depending on the risk-weighted as follows: no risk (in green), potential risk (in orange) and current problem (in red).

## Data Availability

The datasets used and analyzed during the current study will be available from the corresponding author on reasonable request.

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
