# Peer review of "Design, Implementation, and Evaluation of Healthcare Visual Map Tool for Health Workers to Improve Quality of Life of Home Care Patients: Study Protocol"

_healthcare, 2025, doi:10.3390/healthcare13060626_

Round 1
Reviewer 1 Report
Comments and Suggestions for Authors
Dear authors,
Thank you for allowing me to review this article of significant importance which focuses on “Design, implementation and evaluation of Healthcare Visual Map Tool for Health workers to improve the quality of life of home care patients: study protocol”
Abstract
The abstract clearly summarizes the study's aims, methods, and key findings. The use of acronyms should be avoided.
Introduction
The introduction contextualises the problem and the importance of home care well, but could be more concise in some parts, especially when describing ageing statistics. In addition, the transition between the presentation of the problem (fragmentation of information) and the proposed solution (visual map) could be more fluid. Reinforcing the justification for the study, clearly explaining why the existing solutions are not sufficient, would make the introduction more impactful.
Methods
If the study is not yet registered, I suggest considering registration on platforms such as OSF or ClinicalTrials.gov to enhance transparency and credibility.
The article presents a protocol study, but does not mention the use of specific methodological guidelines. To ensure greater transparency and reproducibility. As the study includes a cluster randomised clinical trial, I suggest aligning the methodology of this phase with the SPIRIT Statement, which provides structured recommendations for reporting trial protocols. This will enhance transparency and reproducibility. In addition, if the initial phase includes a systematic review to identify relevant variables, compliance with the PRISMA Statement may be beneficial.
A brief mention of the use of these guidelines in the methods section would strengthen the structure of the manuscript and bring the study into line with international best practice.
The Materials and Methods section is well structured, but some parts could be clearer to ensure greater transparency and reproducibility of the protocol:
- Transition between study phases
In the excerpt: ‘The general framework of this study is shown in Figure 1. In general, the study includes three phases:’
Suggestion: Explain why these three phases were chosen and how each contributes to achieving the study's objective. This will help readers understand the logic of the study design.
- Detailing the statistical analysis
In the excerpt: ‘Outcomes will be compared between groups using the Chi-square test and a multivariate analysis (Cox regression) will be performed introducing adjustment variables into the model: socio-demographic, risk areas and covariates mentioned before.’
Suggestion: Although it is understandable that the full list of covariates has not yet been defined, it would be useful to indicate which criteria will be used to select them (e.g. clinically relevant or statistically significant variables).
- Criteria for excluding participants
In the passage: ‘All patients included in the home care programme will be eligible. They will receive a brief explanation of the study and will be invited to sign the informed consent. Those who do not sign the informed consent will be excluded.’
Suggestion: If there are exclusion criteria envisaged in addition to refusal of consent (e.g. impossibility of follow-up or cognitive barriers), they could be mentioned, even if they are still preliminary.
- Justification of the qualitative analysis
In the excerpt: ‘Data from the interviews and focus groups will be comprehensively systematised with the ATLAS.ti program. The data will be analysed through the thematic content.’
Suggestion: As no data has yet been collected, the authors could justify the choice of thematic analysis over other approaches (e.g. grounded theory), reinforcing its suitability for the study's objective.
These suggestions will help make the protocol more robust and transparent, allowing future readers and reviewers to better understand the methodological decisions.
It is recommended that an Expected Results section be included in the protocol. This section could describe, based on existing literature, what impacts are expected with the implementation of the visual map, such as the reduction of avoidable hospitalisations, improvement in patient safety and optimisation of decision-making by healthcare professionals. This would help to demonstrate the relevance of the study and its potential benefits, without anticipating conclusions.
Discussion
The Discussion section presents well the relevance of the study and the implications of implementing the visual map. However, some improvements could be considered:
Strengthen the link with findings from the literature - Although the text mentions expected benefits, it would be useful to more directly reference previous studies that support these expectations, especially on the effectiveness of similar tools in reducing avoidable hospitalisations.
Explore limitations in greater depth - The article mentions technical challenges in implementing the visual map in the EMR, but could expand the discussion on organisational barriers or resistance from healthcare professionals to adopting the technology.
Clarify the applicability of the findings - In the passage ‘Finally, analysing the effectiveness of the visual map will serve as a proof of concept...’, the statement could be more cautious, reinforcing that the protocol aims to generate initial evidence, but that future studies will be necessary to confirm its effectiveness in clinical practice.
These adjustments would help to strengthen the argument and clarity of the section.
I suggest including a brief Conclusion section to reinforce the importance of the study and highlight next steps. The conclusion could summarise the aim of the protocol, the expected impact of the tool and the relevance of future results for clinical practice. This would help to better structure the closing of the article.
I would like to congratulate the authors on their article. Substantial revisions are required to improve methodological clarity, align with reporting guidelines, and enhance the discussion. These changes will significantly strengthen the manuscript.
Reviewer 2 Report
Comments and Suggestions for Authors
Dear Authors,
Thank you for the opportunity to review this manuscript on such a relevant topic. The study titled "Home Care Visual Map Study" addresses an important area in healthcare. I hope my suggestions will help refine and strengthen the manuscript.
I aim to provide constructive feedback to improve clarity, methodological rigour, and overall coherence of the manuscript. The suggestions are organized by sections, with numbered comments for ease of reference.
1. Abstract
Background (lines 24-26): 1. Consider briefly defining what makes the home environment challenging for patient safety.
Objective (lines 27-29): 2. To avoid confusion, clearly separate the study's aim from the implementation goal.
Methods (lines 30-37): 3.Too detailed for an abstract. Simplify phases 2 and 3 descriptions, focusing on the study design, population, and key outcomes.
Results (lines 38-39): 4. The expected results section is speculative. Consider rephrasing to reflect that these are anticipated outcomes.
Discussion (lines 40-42): 5. Overlaps with the expected results. Focus on potential implications instead
Introduction
Research Question & Objective: 6. The objective is mentioned (lines 131-133), but the research question isn’t explicitly stated. Consider adding a straightforward research question to frame the study.
Contextualization (lines 47-123): 7. Provides strong background information but is relatively lengthy. Streamline by focusing on the most relevant data about frailty, home care, and adverse events.
Gaps in Knowledge: 8. Make the gap in existing literature more explicit to justify the need for this study.
Methodology
-
Study Design: 9. Appropriate for the objectives with clear phases (lines 129-145).
-
Sample Size Calculation (lines 206-209): 10. Adequately calculated, but specify assumptions made (e.g., expected effect size, baseline rates of hospital admissions).
-
Recruitment (lines 210-214): 11. Needs more detail about how patients will be approached and whether there are strategies to minimize bias.
-
Inclusion/Exclusion Criteria: 12. While inclusion is broad (all home care patients), clarify exclusion criteria beyond just informed consent issues.
-
Ethical Aspects: 13. Informed consent is mentioned (line 212), but consider adding details about ethical approval, data protection measures, and confidentiality protocols.
-
Statistical Analysis (lines 233-244): 14. Appropriate overall. However, justify the use of Cox regression in this context—is time-to-event analysis critical for all outcomes? 15. Clarify how missing data will be handled.
4. Results
-
Expression of Results: 16. Since results are not provided yet, ensure that when included, they are concise, with tables and figures supporting key findings. Indicate statistical significance and confidence intervals.
5. Discussion
-
Debate and Interpretation (lines 286-317): 17. The discussion is well-structured but could benefit from a deeper critical analysis comparing the results to existing literature. 18. Consider potential biases and limitations more explicitly, especially regarding the generalizability of the findings.
-
Alignment with Objectives: 19. Ensure the discussion consistently relates to the study objective.
6. Conclusion
-
Adequacy (lines 318-323): 20. Clear and concise. Consider reinforcing the broader implications for healthcare systems beyond the Catalan Institute of Health.
Recommendations:
-
Ensure consistency in terminology (e.g., home care nurses (HCN), home care programs).
-
Proofread for minor grammatical errors and clarity.:
-
Incorrect: The study will be conducted in 20 primary care centres from Tarragona area.
Correction: The study will be conducted in 20 primary care centres in the Tarragona area. -
Incorrect: This phase includes two subphases: identification of the health areas and their contributing factors.
Correction: This phase includes two sub-phases: the identification of health areas and their contributing factors.
"Subphases" should be hyphenated as "sub-phases," and the definite article "the" before "identification" improves clarity. -
Incorrect: Data will be analysed through the thematic content.
Correction: Data will be analysed using thematic content analysis.
The phrase "through the thematic content" is awkward. The correct term is "thematic content analysis." -
Incorrect: A systematic review and meta-analysis confirmed that physical activity significantly improves mobility and activities of daily living in frail elderly people.
Correction: A systematic review and meta-analysis confirmed that physical activity significantly improves mobility and activities of daily living in frail older adults.
The term "elderly people" can be considered outdated; "older adults" is more appropriate in academic writing. -
Incorrect: Home environment has unique characteristics that influence patient safety and the health outcomes.
Correction: The home environment has unique characteristics that influence patient safety and health outcomes.
Missing definite article "the" before "home environment" and unnecessary definite article "the" before "health outcomes."
-
-
Add references where claims are made without direct support, especially regarding statistical data or outcomes from previous studies. Examples:
Original Text (Introduction, lines 49-50):
In 2022, life expectancy at birth in Europe was 80.6 years, whereas in Spain it was 83.2 years.Suggestion:
Add a reference to the source of this statistical data, such as a report from the World Health Organization (WHO), Eurostat, or the National Institute of Statistics (INE) in Spain.Original Text (lines 70-72):
A systematic review has shown that home care programs are associated with lower health services utilization, reduced costs, better quality of life and greater caregiver satisfaction.Suggestion:
Since this statement refers to findings from a systematic review, a specific citation is needed to validate it.Original Text (lines 88-90):
A review including studies from several countries reported AE rates ranging from 3.5% to 15.1%, similar to a Canadian registry indicating an incidence of 4.2%, with 56% preventable.Suggestion:
Cite the review and the Canadian registry directly to support these statistics.Revised Version:
A review including studies from several countries reported AE rates ranging from 3.5% to 15.1% (xxxx) [xx], similar to a Canadian registry indicating an incidence of 4.2%, with 56% preventable (Canadian xxxx, 20xx) [xx].I hope these comments enhance the manuscript. Thank you again for the opportunity to review this important work.
Best regards,
-
Dear Editor,
Thank you for the opportunity to review this manuscript on such a relevant topic. The study, "Home Care Visual Map Study," addresses an important area in healthcare. Based on my review, I believe a major revision must address the key areas outlined below. However, with these improvements, the manuscript has the potential to be an exciting and valuable contribution.
I aim to provide constructive feedback to improve the manuscript's clarity, methodological rigour, and overall coherence. The suggestions are organized by sections, with numbered comments for ease of reference.
Round 2
Reviewer 1 Report
Comments and Suggestions for Authors
Dear Authors,
I sincerely appreciate the revisions you have made, which have significantly enhanced the quality of the article. I believe it now meets the necessary standards for publication. Congratulations to all the authors on this achievement! It has been a pleasure to read your work, and I thank you for the opportunity to do so.
Best regards.
Author Response
Thank you very much for your kind and encouraging words. We are grateful for your thoughtful review and are pleased to hear that the revisions have enhanced the quality of the article.
Reviewer 2 Report
Comments and Suggestions for Authors
Dear Authors,
Thank you for your thoughtful revisions and detailed responses to my comments. I appreciate the effort you have put into addressing the suggestions, and the manuscript has been significantly improved.
Most of the points have been well addressed. However, there are two aspects where further clarification could strengthen the manuscript:
Exclusion Criteria – While the broad inclusion criteria are well justified, a brief clarification on why no additional exclusion criteria were considered (e.g., highly dependent patients, palliative care) would help anticipate potential concerns regarding patient variability.
Overlap Between Results and Discussion – I appreciate your explanation of the distinction between these sections. However, a slight refinement in wording to ensure that expected results remain distinct from the discussion of broader implications might enhance clarity.
These are minor points, and I do not anticipate them being major barriers to acceptance, but addressing them could further strengthen the manuscript.
Thank you again for your work. I look forward to seeing the final version.
Best regards,
Author Response
Exclusion Criteria – While the broad inclusion criteria are well justified, a brief clarification on why no additional exclusion criteria were considered (e.g., highly dependent patients, palliative care) would help anticipate potential concerns regarding patient variability.
Thank you very much for your insightful comment. Regarding the exclusion criteria, we have chosen not to include additional exclusion criteria, as this will be a pragmatic study aimed at representing the reality of the home care population. By adopting this approach, we aim to obtain results that are representative and easily transferable to real-world settings, thus enhancing the external validity and applicability of our findings.
We have added a new sentence in the recruitment section (lines 232-233).
We hope this explanation clarifies our decision, and we are happy to address any further suggestions or concerns you may have.
Overlap Between Results and Discussion – I appreciate your explanation of the distinction between these sections. However, a slight refinement in wording to ensure that expected results remain distinct from the discussion of broader implications might enhance clarity.
Thank you for your valuable comment. To improve clarity, we will refine the wording in the manuscript to more clearly distinguish between the expected results and the discussion of their broader implications. The “Results“ section will focus on the specific outcomes, while the “Discussion” will address their interpretation and impact. We appreciate your suggestion and will incorporate these adjustments to enhance the manuscript.